# Boolean Logic for Low-Energy Deep Learning

**Van Minh Nguyen** [1]   **Cristian Ocampo** [1]   **Aymen Askri** [1]   **Ba-Hien Tran** [1]

## Abstract

Deep learning is computationally intensive. Much effort has been given to reduce the arithmetic complexity whilst energy consumption is the most relevant bottleneck, in which data movement is the dominant part. In addition, the literature focus has been on inference whereas training is several times more intense. In this paper, we make use of the Boolean neuron design and Boolean logic backpropagation to train deep models in the binary domain using Boolean logic instead of gradient descent and real arithmetic. We propose a detailed energy evaluation for both training and inference phases. Our method achieves the best results in standard image classification tasks and consumes almost 27 times less energy with our most efficient and best performing Boolean network. This energy efficiency paves the way for an edge device use, in particular for fine-tuning large models on a dedicated task. In practice, our approach outperforms the state-of-the-art semantic segmentation and shows promising image super-resolution performance.

## 1. Introduction

Running deep models, i.e., *inference*, requires considerable computational resources, it is yet the tip of the iceberg. Deep model *training* is an iterative process involving abundant computation and data for learning. It incurs storing multiple temporal variables and buffers for gradient computation and parameter optimization. This intense process is further repeated for hyper-parameter tuning, running for weeks or months on specialized equipment, resulting in another order of magnitude of carbon footprint and computational resource requirement (Strubell et al., 2019).

The vast majority of works that target the resource con-

---
[1]Mathematical and Algorithmic Sciences Laboratory, Huawei Paris Research Center, France. Correspondence to: Van Minh Nguyen <vanminh.nguyen@huawei.com>.

Accepted to the Workshop on Advancing Neural Network Training at International Conference on Machine Learning (WANT@ICML 2024).

strained training bottleneck focus on the number of arithmetic operations (FLOPS or BOPS)(García-Martín et al., 2019; Qin et al., 2020a) rather than the consumed energy/memory. However, it has been shown that OPs number is meaningless, even harmful, because it does not map directly to actual system complexity. Instead, energy and memory consumption are the consistent and efficient measures of computing hardware (Sze et al., 2017; 2020; Yang et al., 2017; Strubell et al., 2019). In particular, data movement dominates computing in energy consumption and is strictly tied to system architecture, memory hierarchy, and dataflow (Kwon et al., 2019; Sim et al., 2019; Yang et al., 2020a; Chen et al., 2016). Therefore, design effort subjected to reducing OPs *alone* is inefficient. Currently, the main approach to tackle such bottleneck is quantization. It is becoming popular for LLMs (Frantar et al., 2022; Lin et al., 2023; Kim et al., 2023) to enable inference on affordable devices. But only post-training quantization is available on standard GPUs. Better quantized models can be obtained through quantization-aware training (Gupta et al., 2015; Zhang et al., 2018a; Jin et al., 2021; Yamamoto, 2021; Huang et al., 2021; Umuroglu et al., 2017), and quantized training (Chen et al., 2020; Sun et al., 2020; Yang et al., 2022; Chmiel et al., 2021), that reduce the numeric precision of weights, activations, and dataflow from full-precision (FP) to finite-precision format.

A special case of quantization-aware training is binarized neural networks (BNNs) which were first proposed by Courbariaux et al. (2015; 2016) and have been followed by a huge amount of subsequent contributions (Gholami et al., 2022; Zhao et al., 2020; Guo, 2018; Nagel et al., 2021). This design usually binarizes weights and activations to obtain principal forward computation blocks in binary. It learns binarized weights via full-precision latent ones, which are updated by the classical gradient descent backpropagation. The gradient of the binarized variables is usually approximated by a differential proxy of the binarization function, which is most often the identity proxy. Many concurrent approaches (Bai et al., 2018; Ajanthan et al., 2019; 2021; Lin et al., 2020b; Leconte et al., 2023) formulated the BNN learning task as a constrained optimization problem and discussed different methods to generate binary weights from real-valued latent ones. In practice, these works showed that BNNs could achieve state-of-the-art accuracy in study-

level classification problems such as CIFAR-10 or MNIST, but suffer significant accuracy drop on more challenging problems such as ImageNet (Rastegari et al., 2016; Qin et al., 2020a). Besides the reduced network approximation capacity due to lower data precision (Zhou et al., 2016), the use of full-precision optimizers for estimating binary weights are the causes of this degradation. To compensate for this accuracy loss, most recent prominent works (Liu et al., 2020; Nie et al., 2022; Guo et al., 2022) used multiple full-precision components in the network, whereas only a few dataflows have remained binary.

In this work we aim to answer whether energy-friendly deep learning is possible both for inference and training all while maintaining performance. For that, we explore Boolean notions to define networks that are predominantly Boolean, with low energy demands, and that are trained in the binary domain. Extensive experimental evaluation is conducted on a set of common vision tasks requiring moderate to higher levels of accuracy.

Our contributions are: **(1)** We make use of Boolean neuron design and Boolean logic backpropagation principle for the training of deep models in binary domain. **(2)** We introduce a new methodology to fairly assess the energy complexity of both training and inference phases. **(3)** We show competitive or state-of-the-art results of this design in complex computer vision-related tasks, including image super-resolution. **(4)** We show that employing pre-trained Boolean NNs for edge device fine-tuning tasks, such as classification and segmentation, yields very good performances at low energy cost.

## 2. Related works

Energy consumption is a fundamental metric for measuring hardware complexity. However, it requires specific knowledge of computing systems and makes it hard to estimate. Only few results are available, though experimental-based and limited to specific tested models, e.g., Gao et al. (2020); Shao & Brooks (2013); Mei et al. (2014); Bianco et al. (2018); Canziani et al. (2016); García-Martín et al. (2019). Although experimental evaluation is precise, it requires considerable implementation efforts while not generalizing. In addition, most relevant works are only limited to inference and not training (Chen et al., 2016; Kwon et al., 2019; Yang et al., 2020a). Therefore, developing an analytic method to efficiently estimate training energy consumption is desirable.

Regarding NN architectures, significant advances have been made on BNNs (Binarized Neural Networks) for the ImageNet classification task, driving their performance to higher grounds (Guo et al., 2022; Liu et al., 2020; Tu et al., 2022; Lee et al., 2022; Zhang et al., 2022; Xing et al., 2022; Martinez et al., 2020; Wang et al., 2023b). These works, which attempt to reduce the accuracy gap between BNNs and full-precision networks, typically target the primary sources of the computational burden of CNNs, essentially convolutions, data-streams memory (both numeric type and size) and network depth. Consequently, modern BNNs are improved over the following three main areas.

*Binarization strategy.* It seeks to efficiently binarize real-valued data. The sign function is the primary alternative to binarize data-streams, with additional constraints on the data like clipping (Zhang et al., 2022; Guo et al., 2022). ReActNet (Liu et al., 2020) is a prominent work that proposes RSign, a more general alternative to sign, which deals with the fact that distributions may be shifted or biased. A more recent option, Tu et al. (2022) argues that binary values $\{1, -1\}$ might restrain the approximation capabilities of BNNs, which is why they binarize activations to two real values for more representative power.

*Optimization strategy.* Since latent-based training (Courbariaux et al., 2015) remains the underlying method for updating binarized weights, a differential proxy of sign is required. Different or modified alternatives to straight-through-estimator (STE) have been proposed (Liu et al., 2020). Piece-wise polynomials and hyper-parameterized $\tanh$ have been used (Nie et al., 2022). The latent-based approach requires storing both binary and real parameters during training. Furthermore, this approach typically requires a sequential training of multiple stages where activations and weights get progressively converted from full-precision to binary types (Guo et al., 2022; Zhang et al., 2022; Xing et al., 2022), resulting in longer training times. Modern BNN methodologies (Xing et al., 2022; Liu et al., 2020; Guo et al., 2022; Zhang et al., 2022; Tu et al., 2022; Lee et al., 2022; Liu et al., 2022) agree on the fact that using knowledge distillation (KD) closes the gap between BNNs and full-precision models for which additional data augmentation is needed to reduce the common overfitting of BNNs (Guo et al., 2022). In most cases, a single teacher like ResNet34 or ResNet50 suffices to significantly increase the accuracy. More recently, using multi-KD with four teachers, BNext (Guo et al., 2022), reached performances not reported before. In this sense, existing works have investigated BNNs from the perspective that network binarization is considered as a plugin feature to an existing full-precision DNN. For KD, the goal is to transfer the knowledge of a teacher model to a smaller student model. If trained from scratch, the student model generally performs worse than its teacher. However, under the supervision of the teacher network, the binary network can preserve the learning capability and thus obtain comparable performance to the teacher network. Consequently, the process still requires full-precision model training, and cannot tackle the complexity problem of the network training. Furthermore, KD-based training depends on specialized teachers on a particular task, thus reducing

functionality on new data. Helwegen et al. (2019); Wang et al. (2021) proposed some heuristic and improvement of the classic BNN latent-based optimizer.

*Architecture design.* ResNet (Liu et al., 2020; 2018; Guo et al., 2022; Bethge et al., 2020) and MobileNet (Liu et al., 2020; Guo et al., 2022) are the most frequent layouts. In (Zhou et al., 2016; Rastegari et al., 2016) the authors experimented with Alexnet. Among these methodologies, the basic blocks have been greatly transformed. Common modifications include additional shortcuts, automatic channel scaling with Squeeze-and-Excitation (Zhang et al., 2022) or block duplication plus concatenation in the channel domain (Liu et al., 2020; Guo et al., 2022). Recent alternatives incorporate modules that better adapt the input domain to binary dataflows (Xing et al., 2022), replace standard convolutions with lighter pointwise convolutions (Liu et al., 2022), or propose 1-bit alternatives of linear projections (Wang et al., 2023a).

Since the release of ReActNet, the best results are obtained by alternating low-precision dataflows to full-precision after every binary convolution within the network. These works substantially rely on real-valued dataflows during feedforward such as PReLU, Batch Normalization, FP scaling, and further boost accuracy via KD. This highlights the need for native binary neural networks (Nguyen, 2023), and a precise complexity evaluation method to be able to assess gains in regards of memory, energy, and latency.

## 3. Method

### 3.1. Boolean training

The design and training of Boolean layers follows the principle proposed by Nguyen (2023). For illustration purpose, Algorithm 1 presents a pseudo code of our implementation of a Boolean fully-connected layer that uses Boolean logic B *e.g.* XOR, XNOR, or any other. In the forward pass, at iteration $t$, input $x^{l,t}$ is buffered for later use in the backward, and the $j$th neuron output at $k$th sample is computed as:

$$x_{k,j}^{l+1,t} = w_{0,j}^l + \sum_{i=1}^m B\big(x_{k,i}^l, w_{i,j}^l\big), \quad (1)$$

$\forall k \in [1, K], \forall j \in [1, n]$ where $K$, $m$, $n$ are, resp., the training mini-batch, layer input and output size. Here, to take into account the case of real-valued input data, definitions due to Nguyen (2023) are presented in the following. For $x \in \mathbb{R}$, define $x_{\text{bool}} = \text{True} \Leftrightarrow x \geq 0$, and $x_{\text{bool}} = \text{False} \Leftrightarrow x < 0$, and $|x|$ its magnitude. Then, logic operation between $x \in \mathbb{R}$ and $b \in \{\text{False}, \text{True}\}$ is defined according to (Nguyen, 2023) as follows:

$$B(b, x) \stackrel{\text{def}}{=} y, s.t. \ y_{\text{bool}} = B(b, x_{\text{bool}}), \text{ and } |y| = |x|.$$

---

**Algorithm 1:** Illustration of Boolean training with a fully-connected layer.

**Input** : Learning rate $\eta$, nb iterations $T$;

1 **Initialize**
2 $\quad m_{i,j}^{l,0} = 0; \beta^0 = 1;$
3 **for** $t = 0, \ldots, T-1$ **do**
   /* **1. Forward** */
4 $\quad$ Receive and buffer $x^{l,t}$;
5 $\quad$ Compute $x^{l+1,t}$ following (1);
   /* **2. Backward** */
6 $\quad$ Receive $g^{l+1,t}$;
   /* **2.1 Backpropagation** */
7 $\quad$ Compute and backpropagate $g^{l,t}$ following (2);
   /* **2.2 Weight update** */
8 $\quad$ $C_{\text{tot}} := 0, C_{\text{kept}} := 0;$
9 $\quad$ **foreach** $w_{i,j}^l$ **do**
10 $\quad\quad$ Compute $q_{i,j}^{l,t+1}$ following (3);
11 $\quad\quad$ Update $m_{i,j}^{l,t+1} = \beta^t m_{i,j}^{l,t} + \eta^t q_{i,j}^{l,t+1};$
12 $\quad\quad$ $C_{\text{tot}} \leftarrow C_{\text{tot}} + 1;$
13 $\quad\quad$ **if** $\text{XNOR}(m_{i,j}^{l,t+1}, w_{i,j}^{l,t}) = \text{True}$ **then**
14 $\quad\quad\quad$ $w_{i,j}^{l,t+1} \leftarrow \neg w_{i,j}^{l,t};$ /* invert */
15 $\quad\quad\quad$ $m_{i,j}^{l,t+1} \leftarrow 0;$
16 $\quad\quad$ **else**
17 $\quad\quad\quad$ $w_{i,j}^{l,t+1} \leftarrow w_{i,j}^{l,t};$ /* keep */
18 $\quad\quad\quad$ $C_{\text{kept}} \leftarrow C_{\text{kept}} + 1;$
19 $\quad$ Release buffer $x^{l,t}$;
20 $\quad$ Update $\beta^{t+1} \leftarrow C_{\text{kept}}/C_{\text{tot}};$
21 $\quad$ Update $\eta^{t+1};$

---

In the backward pass, this layer receives $g^{l+1,t}$ from downstream layer $l + 1$, which is usually an activation or a batch normalization (BN) layer. The backpropagation through binary activation function is considered for two cases: without *vs.* with BN. In the first case, arithmetic layer is directly followed by the binary activation to avoid BN for complexity reduction, $tanh'$ is used as the backward activation. In the second case, a simple approximation such as the one proposed by (Liu et al., 2018) can be used. Then, backpropagation signal $g^{l,t}$, *cf.* line 7 in Algorithm 1, is computed following Nguyen (2023) as:

$$g_{k,i}^{l,t} = \sum_{j=1}^n \mathbf{1}\big(g_{k,i,j}^{l,t} = \text{True}\big)|g_{k,i,j}^{l,t}|$$

$$- \sum_{j=1}^n \mathbf{1}\big(g_{k,i,j}^{l,t} = \text{False}\big)|g_{k,i,j}^{l,t}|, \quad (2)$$

$\forall k \in [1, K], \forall i \in [1, m]$, where $g_{k,i,j}^{l,t}$ is given according to Nguyen (2023) for the utilized logic B, for example:

$$g_{k,i,j}^{l,t} = \begin{cases} \text{XNOR}(g_{k,j}^{l+1,t}, \neg w_{i,j}^{l,t}), & \text{for XOR neuron,} \\ \text{XNOR}(g_{k,j}^{l+1,t}, w_{i,j}^{l,t}), & \text{for XNOR neuron,} \end{cases}$$

in which $\neg$ is the logic negation. Optimization signal at line

10 in Alogrithm 1 is given according to (Nguyen, 2023) as:

$$q_{i,j}^{l,t+1} = \sum_{k=1}^{K} \mathbf{1}\Big(q_{i,j,k}^{l,t} = \text{True}\Big)|q_{i,j,k}^{l,t}|$$
$$- \sum_{k=1}^{K} \mathbf{1}\Big(q_{i,j,k}^{l,t} = \text{False}\Big)|q_{i,j,k}^{l,t}|, \quad (3)$$

$\forall i \in [1,m], \forall j \in [1,n]$ where

$$q_{i,j,k}^{l,t} = \begin{cases} \text{XNOR}(g_{k,j}^{l+1,t}, \neg x_{k,i}^{l,t}), & \text{for XOR neuron,} \\ \text{XNOR}(g_{k,j}^{l+1,t}, x_{k,i}^{l,t}), & \text{for XNOR neuron.} \end{cases}$$

Finally, the weights are updated in lines 13–18 of Algorithm 1 just following the rule formulated in Nguyen (2023).

## 3.2. Energy estimation

Energy consumption is a fundamental metric for measuring hardware complexity. However, it requires specific knowledge of computing systems and makes it hard to estimate. Few results are available, though experimental-based and limited to specific tested models, *e.g.* (Gao et al., 2020; Shao & Brooks, 2013; Mei et al., 2014; Bianco et al., 2018; Canziani et al., 2016; García-Martín et al., 2019). Although experimental evaluation is precise, it requires considerable implementation efforts while not generalizing. In addition, most relevant works are only limited to inference and not training. (Chen et al., 2016; Kwon et al., 2019; Yang et al., 2020a).

### 3.2.1. HARDWARE SPECIFICATION

In this work, we intend to estimate the training energy consumption on Ascend chip architecture introduced in Liao et al. (2021) and dedicated to DNN computing. Its evaluation on GPU hardware is subject to future work. Ascend architecture has achieved important commercial successes with more than 100 million chips that have been used in real products for applications ranging from smartwatches, smartphones, and smart cars to intelligent clouds.

The core design of Ascend is well described in (Liao et al., 2021). Essentially, it introduces a 3D (cube) computing unit, providing the bulk of high-intensity computation and increasing data reuse. On the other hand, it provides multiple levels of on-chip memory. In particular, memory L0, which is nearest to the computing cube, is tripled to boost further near-memory computing capability, namely L0-A dedicated to the left-hand-side (LHS) input data, L0-B dedicated to RHS input data, and L0-C for the output. For instance, in a convolution, L0-A, L0-B, and L0-C correspond to the input feature maps (ifmaps), filters, and output feature maps (ofmaps), resp. In addition, the output results going through L0-C can be processed by a Vector Unit for in-place operations such as normalization and activation. Table 1 shows

energy efficiency and capacity of the memory hierarchy of a commercial Ascend architecture (Liao et al., 2021).

### 3.2.2. COMPUTE ENERGY

Energy consumption is the sum of compute and memory energies. *Compute energy* is simply given by the number of arithmetic operations multiplied by their unit cost. The number of arithmetic operations is directly determined from the layer's parameters. Their unit cost is obtained by considering the compute efficiency at 1.7 TOPS/W (Liao et al., 2021). For Boolean logic operations, we follow the usual estimation that ADD INT-$n$ costs $(2n - 1)$ logic operations where $n$ stands for bitwidth.

### 3.2.3. MEMORY ENERGY

On the other hand, *memory energy* is all consumed for moving data between their storage through memory levels and the computing unit during the entire lifetime of the process. Since energy consumed at each memory level is given by the number of data accesses to that level times per-access energy cost, it consists in determining the number of accesses to each level of all data streams (*i.e.*, LHS, RHS, Output). Besides taking into account the hardware architecture and memory hierarchy of Ascend chip, our approach to quantifying memory energy is based on existing methods (Chen et al., 2016; Sze et al., 2017; Kwon et al., 2019; Yang et al., 2020a; Horowitz, 2014; Yang et al., 2017) for dataflow and energy evaluation. Given the layer parameters and memory hierarchy, it amounts to ① *Tiling*: determining the tiling strategy for allocating data streams on each memory level; and ② *Data movement*: specifying how data streams are reused or kept stationary to determine their access numbers. In the following, we present our method for the forward and backward passes by taking the example of a convolution layer, as convolutions are the main components of CNNs and the primary source of complexity due to their high data reuse. Denote by $I$, $F$, and $O$ its ifmaps, filters, and ofmaps, respectively.

**Tiling.** Since the ifmaps and filters are usually too large to be stored in buffers, the tiling strategy is aimed at efficiently transferring them to the computing unit. For the forward, denote tiling parameters of ifmaps and filters at memory

| | L3 (DRAM) | L2 | L1 | L0-A | L0-B | L0-C |
|---|---|---|---|---|---|---|
| EE [GBPS/mW] | 0.02 | 0.2 | 0.4 | 4.9 | 3.5 | 5.4 |
| Capacity [KB] | – | | 8192 | 1024 | 64 | 64 | 256 |

Table 1: Memory hierarchy and energy efficiency (EE) of an Ascend core (Liao et al., 2021) used in our evaluation.

level $i$ as follows:

Ifmaps : $[N_i, C_i, H_i^I, W_i^I]$,  Filters : $[M_i, C_i, H^F, W^F]$,

where $H^F \times W^F$ is the filter size, and the remaining notations are used as follows: $N$ –batch size, $M$ –output channels, $C$ –input channels, $H$ –height, and $W$ –width. Tiling is to determine all these parameters, which is an NP-Hard problem (Yang et al., 2020a). The strategy that we follow is to maximize the buffer utilization as well as near compute stationary (*i.e.*, as much reuse as possible to reduce the number of accesses to higher levels). An iterative search over possibilities subject to memory capacity constraint provides tiling combinations of ifmaps and filters on each memory level. The tiling parameters used in this design are listed in the appendix.

**Data movement.** For data movement, at level L0, several data stationary strategies, called *dataflows*, have been proposed in the literature, notably weight, input, output, and row stationary (Chen et al., 2016). Since Ascend chip provides tripled L0 buffers, partial sums can be directly stationary in the computing cube, hence equivalent to output stationary whose implementation is described in (Du et al., 2015). For the remaining levels, our question of interest is how to move ifmaps block $[N_{i+1}, C_{i+1}, H_{i+1}^I, W_{i+1}^I]$ and filters block $[M_{i+1}, C_{i+1}, H^F, W^F]$ from level $i+1$ to level $i$ efficiently. Considering that: (i) ifmaps are reused by the filters over output channels, (ii) filters are reused over the ifmaps spatial dimensions, (iii) filters are reused over the batch dimension, (iv) ifmaps are usually very large whereas filters are small, the strategy that we follow is to keep filters stationary on level $i$ and cycle through ifmaps when fetching them from level $i+1$ as shown in **??** 2. Therein, filters and ifmaps are read block-by-block of their tiling sizes, *i.e.*, filters block $[M_i, C_i, H^F, W^F]$ and ifmaps block $[N_i, C_i, H_i^I, W_i^I]$. Hence, the number of filter accesses to level $i+1$ is 1 whereas the number of ifmaps accesses to level $i+1$ equals the number of level-$i$ filters blocks contained in level $i+1$. Following this method, the number of accesses to memory levels of each data stream can be determined. Hence, denote by $n_i^d$ the number of accesses to level $i$ of data $d$, and $\varepsilon_i$ the energy cost of accessing level $i$, given as the inverse of energy efficiency from Table 1. Following (Chen et al., 2016), the energy cost of moving data $d$ from DRAM (L3) into the cube is given as:

$$\mathcal{E}^d = n_3^d \varepsilon_3 + n_3^d n_2^d \varepsilon_2 + n_3^d n_2^d n_1^d \varepsilon_1 + n_3^d n_2^d n_1^d n_0^d \varepsilon_0. \quad (4)$$

Regarding the output partial sums, the number of accumulations at each level is defined as the number of times each data goes in and out of its lower-cost levels during its lifetime. Its data movement energy is then given as:

$$\mathcal{E}^O = (2n_3^O - 1)\varepsilon_3 + 2n_3^O(n_2^O - 1)\varepsilon_2$$
$$+ 2n_3^O n_2^O(n_1^O - 1)\varepsilon_1 + 2n_3^O n_2^O n_1^O(n_0^O - 1)\varepsilon_0, \quad (5)$$

---

**Algorithm 2:** Data movement from $i+1$ to $i$ levels

**Input:** tiling parameters of ifmaps and filters at levels $i+1$ and $i$.

1 **repeat**
2   read next filters block of size $[M_i, C_i, H^F, W^F]$ from levels $i+1$ to $i$;
3   **repeat**
4     read next ifmaps block of size $[N_i, C_i, H_i^I, W_i^I]$ from levels $i+1$ to $i$;
5     let the data loaded to $i$ be processed;
6   **until** *all ifmaps are read into level $i$*;
7 **until** *all filters are read into level $i$*;

---

where factor of 2 accounts for both reads and writes and the subtraction of 1 is because we have only one write in the beginning (Chen et al., 2016). The appendix provides more details on the number of reuses at each memory level.

**Backward.** For the backward pass, given that $\partial\text{Loss}/\partial O$ is backpropagated from the downstream, it consists in computing $\partial\text{Loss}/\partial F$ and $\partial\text{Loss}/\partial I$. Following the derivation of backpropagation in CNNs by (Zhang, 2016), it is given that:

$$\partial\text{Loss}/\partial F = \text{Conv}(I, \partial\text{Loss}/\partial O), \quad (6)$$
$$\partial\text{Loss}/\partial I = \text{Conv}(\text{Rot}_\pi(F), \partial\text{Loss}/\partial O), \quad (7)$$

where $\text{Rot}_\pi(F)$ is the filter rotated by 180-degree. As a result, the backward computation structure is also convolution operations, hence follows the same process as detailed above for the forward pass.

## 4. Experiments

Our design was benchmarked on different computer vision tasks: classification (using CIFAR-10 (Krizhevsky et al., 2009) and ImageNet (Krizhevsky et al., 2012)) and super-resolution (using DIV2k (Agustsson & Timofte, 2017; Timofte et al., 2017), Set5 (Bevilacqua et al., 2012), Set14 (Zeyde et al., 2012), BSD100 (Huang et al., 2015), and Urban100 (Martin et al., 2001)).

In addition, to conceal that our Boolean Logic is advantageous for edge device learning, we explore the scenario where a pretrained model is deployed to an edge device, *i.e.* fine tuning. On this regard, we analyze two tasks: classification and segmentation. For classification, the trained Boolean VGG-Small architecture is fine-tuned over CIFAR-100. For segmentation, the trained Boolean ResNet18 is used as backbone on DeepLabv3 (Chen et al., 2017) and fine-tuned over the Cityscapes (Cordts et al., 2016), and Pascal VOC 2012 (Everingham et al.) datasets.

In all benchmarks, the Boolean model was built following

| Method | W/A | Acc.(%) | Cons.(%) | Gain (×) |
|---|---|---|---|---|
| Full-precision (Zhang et al., 2018a) | 32/32 | 93.80 | 100.00 | 1.00 |
| BinaryConnect (Courbariaux et al., 2015) | 1/32 | 90.10 | 38.58 | 2.59 |
| XNOR-Net (Rastegari et al., 2016) | 1/1 | 89.83 | 34.21 | 2.92 |
| Hubara *et al*. (Hubara et al., 2017) | 1/1 | 89.85 | 32.60 | 3.06 |
| Boolean w/o BN (Ours) | 1/1 | 90.29 | 3.64 | 27.42 |
| Boolean with BN (Ours) | 1/1 | **92.37** | 4.87 | 20.53 |

Table 2: Experimental results with the standard VGG-Small (ending with 3 FC layers) baseline on CIFAR-10. Energy consumption is evaluated on 1 iteration. 'Cons' and 'Gain' are the energy consumption and gain w.r.t. the FP baseline.

the sketch of the baseline full-precision (FP) architecture such that its arithmetic layers are Boolean and removing FP-specific components, such as ReLU, PReLU activations or BatchNorm (unless mentioned otherwise). Following the literature (Chmiel et al., 2021), the first and the last layers were kept in FP. Adam (Kingma & Ba, 2014) was used as the optimizer of these FP layers, while our Boolean optimizer was used on the remaining Boolean part. The full details of all experiments are provided in the supplementary material.

### 4.1. Image classification

Proof of the proposed concept was initially validated on CIFAR-10 with VGG-Small (Simonyan & Zisserman, 2014) baseline. In the experiments, our boolean architecture follows the layout of (Courbariaux et al., 2015), except that we exclude batch normalization. This configuration obtained a top-1 accuracy of $90.29 \pm 0.09\%$ (estimated over six repetitions), showing similar performance to (Courbariaux et al., 2015), which has 32-bit activations and is full-precision during training (see Table 2). Higher performances are obtained when including batch normalization after convolutions and the activation from (Liu et al., 2018) (referred to as Boolean with BN in the table), with a classification performance equal to $92.37 \pm 0.01\%$ (estimated over five repetitions) which is almost 1 point closer to the FP counterpart. Complementary comparisons with other methodologies using the modified version of VGG-Small (ending with 1 FC layer) are available in the supplementary material.

In terms of energy, it is clear that our Boolean methodology is much more energy efficient for inference and training than latent-based training, which uses FP data streams. Compared to the FP network, our methodology with and without BN achieves $20.53\times$ and $27.42\times$ gain in energy, respectively. Notably, the use of BN provides greater network accuracy at the expense of a slight increase in energy consumption. Yet, even with BN, our methodology provides the best energy/accuracy ratio of all methods.

For ImageNet classification, we use the ResNet18 (He et al., 2015) baseline to validate our Boolean methodology with and without BN. In the former case, our network follows

(Liu et al., 2018) to define the basic block, whilst replacing the average pooling by a Boolean activation on down-sampling blocks, see Figure 7a. The latter case, Boolean w/o BN, uses the Boolean arithmetic components and Boolean activations arranged as the FP analogues in the original down-sampling block (He et al., 2015), see Figure 7b.

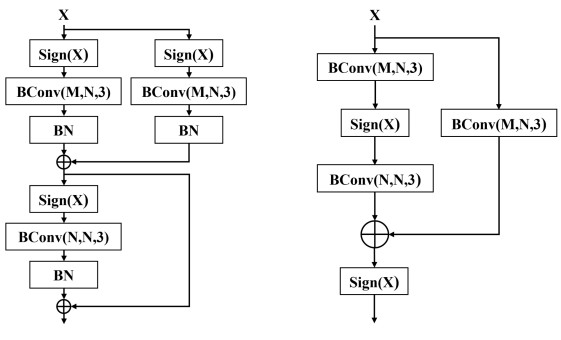

(a) Block with BN.          (b) Block w/o BN.

Figure 1: Proposed Boolean blocks used in the ResNet18 baseline. In the Boolean convolutions, $M$ and $N$ correspond to input and output tensor depth, respectively.

Table 3 presents the results of this experiment. In the comparison, we include methodologies using the standard architecture, and also architectures with additional training strategies to increase performance. For instance, models with larger size, FP data propagation throughout the network via residuals and knowledge distillation (KD)-based learning (with a ResNet34 teacher). In addition to Boolean logic training, it is important to emphasize that our concept has streamlined Boolean architecture. In contrast, the comparative BNNs methods heavily enrich the network with full-precision components, *e.g*., FP shifts (Liu et al., 2020), FP attention weights (Guo et al., 2022) or FP scalars on binary weights (Rastegari et al., 2016) (details are provided in the supplementary material). Even deprived of such FP mechanisms, our native Boolean learning methodology achieves the best accuracy on learning with the basic architecture and without a teacher. In both modalities, our method proves to be the most favorable from the energy point of view, reducing by up to $11.21\times$ the energy consumption of 32-bit FP training and several orders of magnitude better than the popular Bi-RealNet. These findings are paramount for training on low powerful computing devices. Further, based on approximation theory showing that reduced-precision models require a larger size to close the gap to full-precision counterparts (Elbrächter et al., 2021), we investigated how much Boolean models need to be enlarged to reach full-precision performance. Table 3 shows that the Boolean concept recovers totally and outperforms the FP level by $4\times$ filter enlarging, i.e., base 256, at which it still provides $2.58\times$ energy reduction. It also outperforms PokeBNN (Zhang et al., 2022), which uses ResNet50 as a teacher. In order to avoid additional computational burden during evaluation, the log-

its predicted from the ResNet34 teacher were precomputed before launching the experiments.

| Training Modality | Method | Acc. (%) | Cons. (%) | Gain (×) |
|---|---|---|---|---|
| Baseline | ResNet18 (He et al., 2015) | 69.7 | 100.00 | 1.00 |
| | BNN (Courbariaux et al., 2016) | 42.2 | – – | – – |
| | XNOR-Net (Rastegari et al., 2016) | 51.2 | – – | – – |
| | **Ours with BN (Base 64)** | 51.8 | 8.92 | 11.21 |
| FP Shortcut | Bi-RealNet:18 (Liu et al., 2018) | 56.4 | 46.60 | 2.15 |
| Larger Models | Bi-RealNet:34 (Liu et al., 2018) | 62.2 | 80.00 | 1.25 |
| | Bi-RealNet:152 (Liu et al., 2018) | 64.5 | – – | – – |
| | Melius-Net:29 (Bethge et al., 2020) | 65.8 | – – | – – |
| | **Ours w/o BN (Base 256)** | **66.9** | 38.82 | 2.58 |
| KD: ResNet34 | Real2Binary (Martinez et al., 2020) | 65.4 | – – | – – |
| | ReActNet-ResNet18 (Liu et al., 2020) | 65.5 | 45.43 | 2.20 |
| | BNext:18 (Guo et al., 2022) | 68.4 | 47.48 | 2.11 |
| | Ours with BN (Base 192) | 65.9 | 26.91 | 3.72 |
| | **Ours w/o BN (Base 256)** | **70.0** | 38.82 | 2.58 |
| KD: ResNet50 | PokeBNN-ResNet18 (Zhang et al., 2022) | 65.2 | – – | – – |

Table 3: ImageNet classification performance for multiple binary methodologies using different training settings on the ResNet18 baseline. Energy consumption is evaluated on 1 iteration. 'Base' refers to the output channel of the first convolution. 'Cons' and 'Gain' are the energy consumption and gain w.r.t. the FP baseline.

For completeness, we also implemented neural gradient quantization to quantize it by using INT4 quantization with logarithmic round-to-nearest approach (Chmiel et al., 2021) and statistics aware weight binning (Choi et al., 2018). On ImageNet, we confirm that 4 bits quantization is enough to recover standard backpropagation performances (67.53% in 100 epochs, more details in the appendix).

## 4.2. Fine-tuning

The detailed energy analysis from Section 3.2 indicates that our method consumes significantly less energy than other popular BNN methodologies (see Table 2). This suggests that Boolean trained networks are particularly well suited for training and fine-tuning on low capacity devices. The main idea is to take advantage of a large central server that is amenable to gather large number of samples for training a Boolean NN with low error. This pre-trained network is then downloaded by edge devices. We assume edge devices have limited computational capacity, and limited access to data. Hence, our goal is to employ a Boolean backbone, and fine-tune it locally (*i.e.* directly on-device).

### 4.2.1. BOOLEAN FINE-TUNING FOR CLASSIFICATION

For this experiment, we aim at exploring the adaptation capabilities of our methodology to similar problems but different data. We used the VGG-Small architecture trained on CIFAR-10 and fine-tune to CIFAR-100. We also show the results when the starting model was trained on CIFAR-100 and fine-tuned to CIFAR-10. In all experiments the results are provided without BN on the architectures.

| Ref. | Method | Model Init. | Train./FT Dataset | Bitwidth W/A/G | Acc. (%) |
|---|---|---|---|---|---|
| A | Baseline FP | Random | CIFAR-10 | 32/32/32 | 95.27 |
| B | Baseline FP [1] | Random | CIFAR-100 | 32/32/32 | 77.27 |
| C | **Ours** | Random | CIFAR-10 | 1/1/16 | 90.29 |
| D | **Ours**[1] | Random | CIFAR-100 | 1/1/16 | 68.43 |
| E | Baseline FP[1] | A | CIFAR-100 | 32/32/32 | 76.74 |
| F | **Ours**[1] | C | CIFAR-100 | 1/1/16 | 68.37 |
| G | Baseline FP | B | CIFAR-10 | 32/32/32 | 95.77 |
| H | **Ours** | D | CIFAR-10 | 1/1/16 | 92.09 |

Table 4: Top-1 accuracy of the proposed Boolean methodology with the VGG-Small architecture fine-tuned on CIFAR-10 and CIFAR-100. 'FT' means 'Fine-Tuning'.

Table 4 shows the set of fine-tuning experiments performed on VGG-Small. For all our experiments, including the reproduction of the FP baseline, we used during training: random horizontal flip and mixup learning (with random alpha blending factor up to 0.2). Notice that fine-tuning our trained Boolean model on CIFAR-100 (Ref. F) yields a model that is almost identical to the Boolean model that was entirely trained from scratch (Ref. D). Even more outstanding is the case when the Boolean model is fine-tuned on CIFAR-10 (Ref. H), with the final prediction accuracy being 1.8% higher than the Boolean model trained from random initialization (Ref. C).

These results indicate that our methodology adequately captures the underlying information needed to characterize an object, in the sense of classification. In particular, when fine-tuning to CIFAR-10 it takes advantage of its initial spectrum of learned categories to better adjust the weights to new classes. In this case, even from the first epoch it gets 80.91%, surpassing 90% of the final accuracy of the reference model (Ref. C). Correspondingly, fine-tuning to CIFAR-100 requires considerably more training to reach 90% of the final accuracy of the reference model (Ref. D). This is mainly because the number of categories is 10 times larger with less labels per class, see Figure 2. It is worth noting that, as in the FP models, the fine tuning of our Boolean models also ensures robust learning when the initial model was trained on a more complex task.

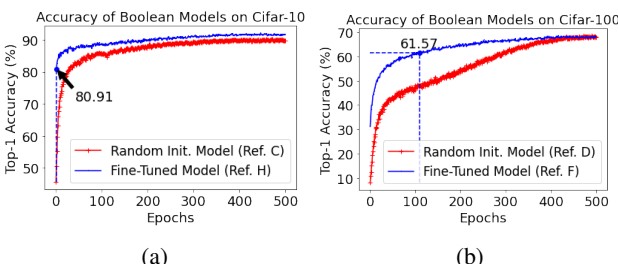

(a)          (b)

Figure 2: Accuracy curves during fine-tuning (blue & thin line) and number of epochs required to reach 90% of the final accuracy of reference models (red & thick line).

[1]VGG-Small with the last FC layer mapping to 100 classes.

### 4.2.2. BOOLEAN FINE-TUNING FOR SEGMENTATION

We have extended the application of our novel Boolean Logic training methodology to the realm of semantic segmentation, a task that necessitates pixel-level classification and thus requires highly detailed feature extraction. This is a formidable challenge for highly quantized networks. Leveraging the success of our method in image classification, we adapted the same network architecture, to cater to the intricacies of semantic segmentation. For this experiment the baseline is the DeepLabv3 (Chen et al., 2017) network structure using as backbone the pretrained Boolean ResNet18 without BN (Figure 7b) and followed by the Boolean Atrous Pyramid Pooling (BoolASPP) module. Our experimental findings demonstrate that this approach not only retains the inherent lightweight advantages of extremely quantized NNs, but also markedly improves the performance in complex semantic segmentation tasks.

We utilized the AdamW (Loshchilov & Hutter, 2017) optimizer with an initial learning rate of $5 \times 10^{-4}$ for real parameters, and the Boolean Logic optimizer with an initial learning rate of 12 for Boolean parameters. Given the distinct characteristics of Boolean logic learning, particularly in the early stages of training, we noticed a tendency for parameters to flip easily due to large backward signals in semantic segmentation tasks. This issue is further exacerbated by the fact that each pixel in the image contributes to the backward signal, even in small batch sizes. To better preserve the integrity of the pretrained backbone, we reduced the Boolean learning rate within the backbone from 12 to 6. Throughout the process, we employed a polynomial learning rate policy with a power factor of $p = 0.9$ for all parameters and conducted the optimization using the cross-entropy loss function. The model was trained on Cityscapes (Cordts et al., 2016) dataset for $140K$ iterations, or on Pascal VOC 2012 dataset (Everingham et al.) for $80K$ iterations, with a batch size of 8. It is also noteworthy that we refrained from using auxiliary loss or knowledge distillation techniques, as these methods introduce additional computational burdens, which contradict our goal of efficient on-device training.

As demonstrated in Table 5, our proposed method attains a mIoU of 67.4% on the Cityscapes dataset. This performance significantly surpasses previous BNN attempts and closely approaches the efficacy of full-precision networks. Similarly, our methodology yields results close to the FP baseline on the Pascal VOC 2012 dataset. Notably, this enhancement is achieved without necessitating the intermediary use of floating-point parameters during the training process, underscoring the efficiency and efficacy of our approach.

### 4.3. Image super-resolution

We evaluate our Boolean design capabilities to synthesize data. In order to capitalize efficiency, we used the small

| Dataset | Model | mIoU (%) |
|---|---|---|
| | FP baseline | 70.7 |
| Cityscapes | Binary DAD-Net (Frickenstein et al., 2020) | 58.1 |
| | **Ours** | **67.4** |
| Pascal VOC 2012 | FP baseline | 72.1 |
| | **Ours** | 67.3 |

Table 5: Performance on image segmentation tasks.

version of the popular EDSR method (Lim et al., 2017) for super-resolution, with eight residual blocks, later referred to as *Small EDSR*. Our Boolean architecture uses Boolean residual blocks as the proposed for image classification, see Figure 7b. The results with the modified FP counterpart were obtained using the official implementation[2].

Table 6 summarizes the results of our experiments. The proposed Boolean Logic methodology obtains values very close to those of the FP reference at each task. It shows the best results for datasets like Set14 and BSD100. Notoriously, as is the case for the reference method, the Boolean methodology suffers significant performance reduction when the desired scaling factor is $\times 4$. Notice that on high-resolution images like Div2K, our method generates images with high PSNR, even higher on low-resolution images like Set5. These results demonstrate that our Boolean methodology can perform pretty well on detail-demanding tasks while being considerably robust to image resolution.

| Task | Method | Set5 | Set14 | BSD100 | Urban100 | Div2K |
|---|---|---|---|---|---|---|
| | Full EDSR (FP) | 38.11 | 33.92 | 32.32 | 32.93 | 35.03 |
| $\times 2$ | Small EDSR (FP) | 38.01 | 33.63 | 32.19 | 31.60 | 34.67 |
| | Ours | 37.42 | 33.00 | 31.75 | 30.26 | 33.82 |
| | Full EDSR (FP) | 34.65 | 30.52 | 29.25 | —— | 31.26 |
| $\times 3$ | Small EDSR (FP) | 34.37 | 30.24 | 29.10 | —— | 30.93 |
| | Ours | 33.56 | 29.70 | 28.72 | —— | 30.22 |
| | Full EDSR (FP) | 32.46 | 28.80 | 27.71 | 26.64 | 29.25 |
| $\times 4$ | Small EDSR (FP) | 32.17 | 28.53 | 27.62 | 26.14 | 29.04 |
| | Ours | 31.23 | 27.97 | 27.24 | 25.12 | 28.36 |

Table 6: PSNR (dB) Performance of the proposed Boolean methodology for super-resolution using the EDSR baseline.

## 5. Conclusion

We have presented a method for training deep neural networks that is provably efficient for resource-constrained environments. In particular, we have developed a method to estimate the energy consumption of NN training and apply it to our Boolean architectures. Our results suggest that full-precision performance can be totally recovered by enlarged Boolean models while gaining multifold complexity reduction. One can fine-tune these energy-efficient models on edge devices for specific tasks. Our experiments highlight that Boolean models can handle finer tasks, contrary to the misbelief that binary models only work for image classification.

---

[2]https://github.com/sanghyun-son/EDSR-PyTorch

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

# Boolean Logic for Low-Energy Deep Learning

## Supplementary Material

## A. Memory energy

This section provides supplementary details on how the number of accesses to each memory level in Ascend chip architecture is determined by taking as example a 2D convolution layer whose parameters are summarized in Table 7.

### A.1. Forward

In the forward pass, there are three types of input data reuse:

- For an $H^I \times W^I$ ifmap, there are $H^O \times W^O$ convolutions performed with a single $H^F \times W^F$ filter to generate a partial sum. The filter is reused $H^O \times W^O$ times, and this type of reuse is defined as *filter convolutional reuse*. Also, each feature in the ifmaps is reused $H^F \times W^F$ times, and this is called *feature convolutional reuse*.

- Each ifmap is further reused across $M$ filters to generate $M$ output channels. This is called *ifmaps reuse*.

- Each filter is further reused across the batch of $N$ ifmaps. This type of reuse is called *filter reuse*.

As mentioned in Section 3.2, a loop tiling strategy for convolutional layers is necessary in order to transmit ifmaps and filters through the memory hierarchy efficiently. Determining tiling parameters, which are summarized in Table 8, is a combinatorial problem. Different approaches can be used and **??** 3 shows an example that explores the best tiling parameters subjected to maximizing the buffer utilization and near compute stationary. Therein, the amount of data stored in level $i$ is

| Parameter | Description |
|:---:|:---|
| $N$ | batch size |
| $M$ | number of ofmaps channels |
| $C$ | number of ifmaps channels |
| $H^I/W^I$ | ifmaps plane height/width |
| $H^F/W^F$ | filters plane height/width |
| $H^O/W^O$ | ofmaps plane height/width |

Table 7: Shape parameters of a convolution layer.

| Parameter | Description |
|:---:|:---|
| $M_2$ | number of tiling weights in L2 buffer |
| $M_1$ | number of tiling weights in L1 buffer |
| $M_0$ | number of tiling weights in L0-B buffer |
| $N_2$ | number of tiling ifmaps in L2 buffer |
| $N_1$ | number of tiling ifmaps in L1 buffer |
| $N_0$ | number of tiling ifmaps in L0-A buffer |
| $H_2^I/W_2^I$ | height/width of tiling ifmaps in L2 buffer |
| $H_1^I/W_1^I$ | height/width of tiling ifmaps in L2 buffer |
| $H_0^I/W_0^I$ | height/width of tiling ifmaps in L0-A buffer |

Table 8: Tiling parameters of a convolution layer.

---

**Algorithm 3:** Loop tiling strategy in the $i$th level

---

**Input:** tiling parameters of ifmaps and filters at level $i + 1$, and buffer capacity of level $i$.
**Output:** tiling parameters of ifmaps and filters at level $i$.

1 **Initialize**
2    $\mathcal{E}^{\min} := \infty$;
3 **for** $M_i \leftarrow M_{i+1}$ *to* $1$ **do**
4    **for** $N_i \leftarrow N_{i+1}$ *to* $1$ **do**
5      **for** $H_i^I \leftarrow H_{i+1}^I$ *to* $H^F$ **do**
6        **for** $W_i^I \leftarrow W_{i+1}^I$ *to* $W^F$ **do**
7          Calculate $Q_i$, the required amount of ifmaps and filters to be stored in the $i$th level of capacity $Q_i^{\max}$;
8          Calculate $\mathcal{E}_i$, the energy cost of moving ifmaps and filters from the $i$th level;
9          **if** *($Q_i \leq Q_i^{max}$) and ($\mathcal{E}_i < \mathcal{E}^{min}$)* **then**
10            Retain tiling parameters as best;
11            $\mathcal{E}^{\min} \leftarrow \mathcal{E}_i$;
12 **return** *Best tiling parameters*

---

| Data | DRAM (L3) | L2 | L1 | L0 |
|---|---|---|---|---|
| $I$ ($n_i^I$) | $\left\lceil \frac{M}{M_2} \right\rceil \times \frac{\alpha^v}{\alpha_2^v} \times \frac{\alpha^h}{\alpha_2^h}$ | $\left\lceil \frac{M_2}{M_1} \right\rceil \times \frac{\alpha_2^v}{\alpha_1^v} \times \frac{\alpha_2^h}{\alpha_1^h}$ | $\left\lceil \frac{M_1}{M_0} \right\rceil \times \frac{\alpha_1^v}{\alpha_0^v} \times \frac{\alpha_1^h}{\alpha_0^h}$ | $H^F \times W^F \times \alpha_0^v \times \alpha_0^h$ |
| $F$ ($n_i^F$) | $1$ | $\left\lceil \frac{N}{N_2} \right\rceil \times \left\lceil \frac{H^O}{H_2^O} \right\rceil \times \left\lceil \frac{W^O}{W_2^O} \right\rceil$ | $\left\lceil \frac{N_2}{N_1} \right\rceil \times \left\lceil \frac{H_2^O}{H_1^O} \right\rceil \times \left\lceil \frac{W_2^O}{W_1^O} \right\rceil$ | $\left\lceil \frac{N_1}{N_0} \right\rceil \times \left\lceil \frac{H_1^O}{H_0^O} \right\rceil \times \left\lceil \frac{W_1^O}{W_0^O} \right\rceil$ |
| $O$ ($n_i^O$) | $1$ | $1$ | $1$ | $1$ |

Table 9: Numbers of accesses at different memory levels of forward convolution.

calculated as:

$$Q_i^I = N_i \times C_i \times H_i^I \times W_i^I \times b^I,$$
$$Q_i^F = M_i \times C_i \times H^F \times W^F \times b^F, \tag{8}$$

where $Q_i^I / Q_i^F$ and $b^I / b^F$ represent the memory and bitwidth of ifmaps/filters, respectively. From the obtained tiling parameters, the number of accesses that is used for (4) and (5) is determined by taking into account the data movement strategy as shown in **??** 2. As a result, Table 9 summarizes the number of accesses to memory levels for each data type in the forward pass. Therein, $\alpha^v = H^O/H^I$, $\alpha^h = W^O/W^I$, $H_i^O/W_i^O$ define the height/width of tiling ofmaps in L$i$ buffers, $\alpha_i^v = H_i^O/H_i^I$, and $\alpha_i^h = W_i^O/W_i^I$ for $i = 2, 1$, and $0$.

### A.2. Backward

In the backward pass, it consists in computing two gradients $\partial \text{Loss}/\partial F$ and $\partial \text{Loss}/\partial I$. As described in Section 3.2, we can evaluate the memory energy of the backward pass by following the exact mechanism of the forward pass with the respective shape parameters. For instance, Table 10 summarizes the number of accesses at each memory level in the backward pass when calculating the gradient $G^I = \partial \text{Loss}/\partial I$. Therein, $C_i$ defines the number of tiling ifmaps in L$i$ buffer, $\beta^v = H^I/H^O$, $\beta^h = W^I/W^O$, $\beta_i^v = H_i^I/H_i^O$, and $\beta_i^h = W_i^I/W_i^O$ for $i = 2, 1$, and $0$.

## B. Experimental design

### B.1. Training setup

The presented methodology and the architecture of the described Boolean NNs were implemented in Pytorch and trained on Nvidia GPUs Tesla V100. The networks thought predominantly Boolean, also contain a fraction of FP parameters that were optimized using the Adam optimizer with learning rate $10^{-3}$. For learning the Boolean parameters we used the Boolean optimizer. Training the Boolean networks for Image Classification was conducted with learning rates $\eta = 150$ and $\eta = 12$,

| Data | DRAM (L3) | L2 | L1 | L0 |
|---|---|---|---|---|
| $O\ (n_i^O)$ | $\lceil\frac{C}{C_2}\rceil \times \frac{\beta^v}{\beta_2^v} \times \frac{\beta^h}{\beta_2^h}$ | $\lceil\frac{C_2}{C_1}\rceil \times \frac{\beta_2^v}{\beta_1^v} \times \frac{\beta_2^h}{\beta_1^h}$ | $\lceil\frac{C_1}{C_0}\rceil \times \frac{\beta_1^v}{\beta_0^v} \times \frac{\beta_1^h}{\beta_0^h}$ | $H^F \times W^F \times \beta_0^v \times \beta_0^h$ |
| $F\ (n_i^F)$ | $1$ | $\lceil\frac{N}{N_2}\rceil \times \lceil\frac{H^I}{H_2^I}\rceil \times \lceil\frac{W^I}{W_2^I}\rceil$ | $\lceil\frac{N_2}{N_1}\rceil \times \lceil\frac{H_2^I}{H_1^I}\rceil \times \lceil\frac{W_2^I}{W_1^I}\rceil$ | $\lceil\frac{N_1}{N_0}\rceil \times \lceil\frac{H_1^I}{H_0^I}\rceil \times \lceil\frac{W_1^I}{W_0^I}\rceil$ |
| $G^I\ (n_i^{G^I})$ | $1$ | $1$ | $1$ | $1$ |

Table 10: Numbers of accesses at different memory levels for $\partial\mathrm{Loss}/\partial I$.

for architectures with and without batch normalization, respectively. During the experiments, both optimizers used the cosine scheduler iterating over 300 epochs.

We highlight the importance of using data augmentation techniques when training low bit-width models which otherwise would overfit with standard techniques. In addition to techniques like random resize crop or random horizontal flip, we used RandAugment, lighting (Liu et al., 2020) and Mixup (Zhang et al., 2018b). Following (Touvron et al., 2019), we used different resolutions for the training and validation sets. For ImageNet, the training images were 192×192 px and 224×224 px for validation images. The batch size was 300 for both sets and the cross-entropy loss was used during training.

### B.2. CIFAR-10

VGG-Small is found in the literature with different fully-connected FC layers. Several works take inspiration from the classic work of (Courbariaux et al., 2015), which uses 3 FC layers. Since other BNN methodologies only use a single FC layer, Table 11 presents the results with the modified VGG-Small.

| Method | Forward Bit-width (W/A) | Training Bit-width (W/G) | Acc. (%) |
|---|---|---|---|
| FP | 32/32 | 32/32 | 93.8 |
| XNor-Net (Rastegari et al., 2016) | 1/1 | 32/32 | 87.4 |
| LAB (Hou et al., 2016) | 1/1 | 32/32 | 87.7 |
| RAD (Ding et al., 2019) | 1/1 | 32/32 | 90.0 |
| IR-Net (Qin et al., 2020b) | 1/1 | 32/32 | 90.4 |
| RBNN (Lin et al., 2020a) | 1/1 | 32/32 | 91.3 |
| SLB (Yang et al., 2020b) | 1/1 | 32/32 | 92.0 |
| Ours | 1/1 | 1/16 | 90.8 |

Table 11: Top-1 accuracy for different binary methodologies using the modified VGG-Small (ending with 1 FC layer) on the CIFAR-10 dataset.

### B.3. Ablation study on image classification

The final block design for image classification was established after iterating over two models. The Boolean blocks examined were evaluated using the ResNet18 baseline architecture and adjusting the training settings to improve performance. Figure 7 presents the preliminary designs.

The Boolean Block I, Figure 3a, is similar to the original ResNet18 block in that BN operations are removed and ReLUs are replaced by the Boolean activation. This design always includes a convolution in the shortcut with spatial resolution being handled by the stride. Notice that for this block we add a Boolean activation after the Maxpool module in the baseline (also for the final baseline architecture). The Boolean Block II, Figure 3b, is composed by two stacked residual modules. For downsampling blocks we use the reshaping operation to reduce the spatial resolution and enlarge the channel dimensions both by a factor of 2. The shortcut is modified accordingly with different operations in order to guarantee similar spatial dimensions before the summation.

Table 12 summarizes the results obtained with the proposed designs on ImageNet. During our experimentation, we validated the hypothesis that increasing network capacity on the convolutional layers yielded higher accuracy values. However, similar to FP CNNs, we confirmed there is a limit by which the hypothesis ceases to be true, leading to overfitting. Incorporating a

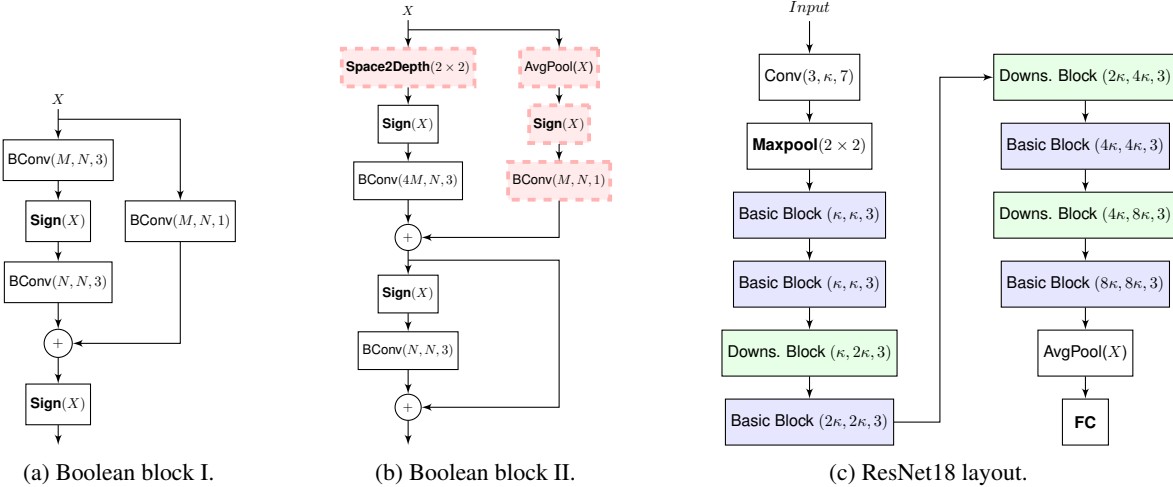

|     |     |     |
| :-: | :-: | :-: |
| (a) Boolean block I. | (b) Boolean block II. | (c) ResNet18 layout. |

Figure 3: Preliminary designs for the baseline architecture and the Boolean basic blocks. The dashed and red-shaded operations in the Boolean block II are introduced for downsampling blocks.

more severe training strategy had a sustained positive impact. Even so, for larger configurations, the compromise between accuracy and size can be cumbersome.

Among the strategies to reduce overfitting during training we included: mixup data-augmentation (Zhang et al., 2018b), image illumination tweaking, rand-augment and smaller input resolution for training than for validation (Touvron et al., 2019). All combined, increased the accuracy by ∼3 points (check results for Block II + base channel 230 with and w/o additional data augmentation).

Compared to Block II, notice that the data streams in Block I are predominantly Boolean throughout the design. This is because it makes use of lightweight data types such as integer (after convolutions) and binary (after activations). In addition, it avoids the need of using a spatial transformation that may affect the data type and data distribution. In that regard, Block II requires 4 times more parameters for the convolution after reshaping, than the corresponding operation in Block I. This is exacerbated in upper layer convolutions, where the feature maps are deeper. Therefore, it makes sense to use Block I, as it is lighter and less prone to overfitting when the network capacity is expanded.

### B.4. Image super-resolution

The seminal EDSR (Lim et al., 2017) method for super-resolution was used together with our Boolean methodology. In particular, the residual blocks are directly replaced by our Boolean basic block, see Figure 4. For all three tasks in super-resolution, training was carried out with small patches of 96×96 px (40 of them extracted randomly from each single image in the Div2K dataset) and validated with the original full-resolution images. The learning rate for real and boolean

| Block Design | Base Channel | $1^{st}$ Conv. Bit-width | Shortcut Fil. Size | Data Augmentation | Acc. (%) |
| :---: | :---: | :---: | :---: | :---: | :---: |
| Block I | 128 | 32 | $1 \times 1$ | Random Crop, Random Flip | 53.35 |
|  | 192 | 32 | $1 \times 1$ | Random Crop, Random Flip | 56.79 |
|  | 192 | 32 | $1 \times 1$ | Lighting, Mixup, RandAugment and (Touvron et al., 2019) | 61.90 |
|  | 256 | 32 | $1 \times 1$ | Lighting, Mixup, RandAugment and (Touvron et al., 2019) | 64.32 |
|  | 256 | 32 | $3 \times 3$ | Lighting, Mixup, RandAugment and (Touvron et al., 2019) | **66.89** |
| Block II | 128 | 1 | $1 \times 1$ | Random Crop, Random Flip | 56.05 |
|  | 128 | 32 | $1 \times 1$ | Random Crop, Random Flip | 58.38 |
|  | 192 | 32 | $1 \times 1$ | Random Crop, Random Flip | 61.10 |
|  | 192 | 32 | $1 \times 1$ | Lighting, Mixup, RandAugment and (Touvron et al., 2019) | 63.21 |
|  | 230 | 32 | $1 \times 1$ | Random Crop, Random Flip | 61.22 |
|  | 230 | 32 | $1 \times 1$ | Lighting, Mixup, RandAugment and (Touvron et al., 2019) | **64.41** |

Table 12: Evaluation of the proposed blocks in ImageNet and their respective configurations during training.

parameters were $10^{-4}$ and $\eta = 36$, respectively. The networks were trained by minimizing the $L_1$-norm between the ground-truth and the predicted upsampled image while using the Adam optimizer and Boolean optimizer. In our experiments the batch size was 20. Some example images generated by our methodology are showed in Figures 5 and 6.

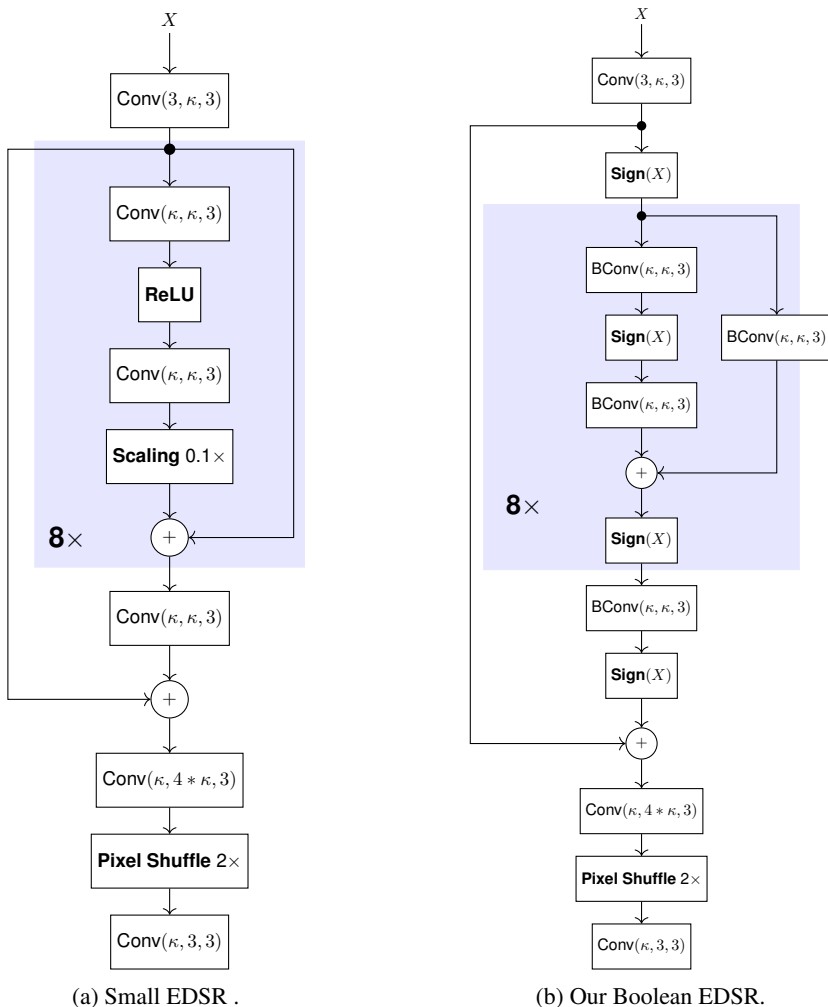

(a) Small EDSR .                    (b) Our Boolean EDSR.

Figure 4: Small EDSR for single scale $\times 2$ super-resolution and our Boolean version with Boolean residual blocks. In both architectures the channels dimensions are $\kappa = 256$ and the shaded blocks are repeated $8\times$.

### B.5. Neural gradient quantization

In the backward pass we implement, only the backpropagation signal is not Boolean when diff. majority aggregation is used (Nguyen, 2023). Thus, for completeness, we also implemented neural gradient quantization to quantize it by using INT4 quantization with logarithmic round-to-nearest approach (Chmiel et al., 2021) and statistics aware weight binning (Choi et al., 2018). Statistics aware weight binning is a method that seeks for the optimal scaling factor, per layer, that minimizes the quantization error based on the statistical characteristics of neural gradients. It involves per layer additional computational computations, but stays negligible with respect to other (convolution) operations. On ImageNet, we recover the findings from (Chmiel et al., 2021): 4 bits quantization is enough to recover standard backpropagation performances.

### B.6. Popular binary basic blocks for classification

Recent BNNs methodologies have proposed different mechanisms to improve performance. Most of them exploit full-precision operations to adjust datastreams within the network, like shift and scaling factors before binary activations (Liu

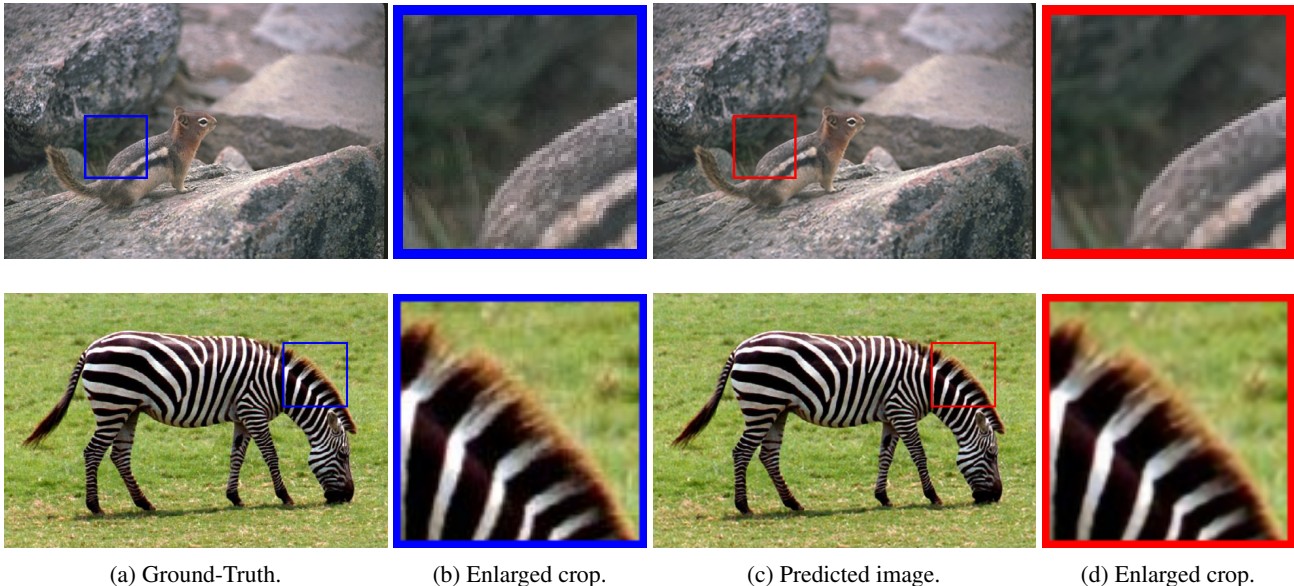

| (a) Ground-Truth. | (b) Enlarged crop. | (c) Predicted image. | (d) Enlarged crop. |

Figure 5: Ground-truth high resolution images and the output of our Boolean super-resolution methodology. First row: image "013" from BSD100, with PSNR: 35.54 dB. Second row: image "014" from Set14, with PSNR: 33.92 dB.

et al., 2020) or channel scaling through Squeeze-and-Excitation modules (Martinez et al., 2020; Guo et al., 2022). Figure 7 shows the basic blocks of three methodologies that perform particularly well in ImageNet. Together with BN and regular activations, those techniques not only add an additional level of complexity but also lead to heavier use of computational resources and latency delays.

For comparison we also show the proposed block (Figure 7a) used in our experiments for Image Classification, Image Segmentation and Image Super-Resolution. Our block is compact in the sense that it only includes Boolean convolutions and Boolean activations, strategically placed to keep the input and output datastreams Boolean.

## B.7. Semantic segmentation

### B.7.1. NETWORK ARCHITECTURE

Our Boolean architecture is based on DeepLabv3 (Chen et al., 2017), which has shown great success in semantic segmentation. It is proven that using dilated or atrous convolutions, which preserve the large feature maps, instead of strided convolutions is prominent for this task. In our Boolean model with ResNet-18 layout, we replace the strided convolutions in the last two ResNet layers with the non-strided version, and the dilated convolutions are employed to compensate for the reduced receptive field. Thus, the images are $8\times$ downsampled instead of $32\times$, preserving small object features and allowing more information flow through the Boolean network. As shown in Figure 7, in the Boolean basic block, a $3 \times 3$ convolution instead of $1 \times 1$ convolution is used to ensure the comparable dynamic range of pre-activations between the main pass and the short-cut. Keeping these Boolean convolutional layers non-dilated naturally allows the backbone to extract multi-scale features without introducing additional computational cost.

The Atrous Spatial Pyramid Pooling (ASPP) consists of multiple dilated convolution layers with different dilation rates and global average pooling in parallel, which effectively captures multi-scale information. In the Boolean ASPP (BoolASPP), we use one $1 \times 1$ Boolean convolution and three $3 \times 3$ Boolean dilated convolution with dilation rates of $\{12, 24, 36\}$ following by Boolean activation functions. The global average pooling (GAP) branch in ASPP captures image-level features, which is crucial for global image understanding as well as large object segmenting accuracy. However, in BoolASPP, as shown in Figure 9c, the Boolean input $X$ leads to significant information loss before the global average pooling may cause performance degradation on large objects. Therefore, we keep the inputs integer for the GAP branch as demonstrated in Figure 9d. To prevent numerical instability, batch normalization is used in the GAP branch before each activation function. Using BoolASPP enhances the multi-scale feature extraction and avoids parameterized upsampling layers, *e.g.* transposed convolution.

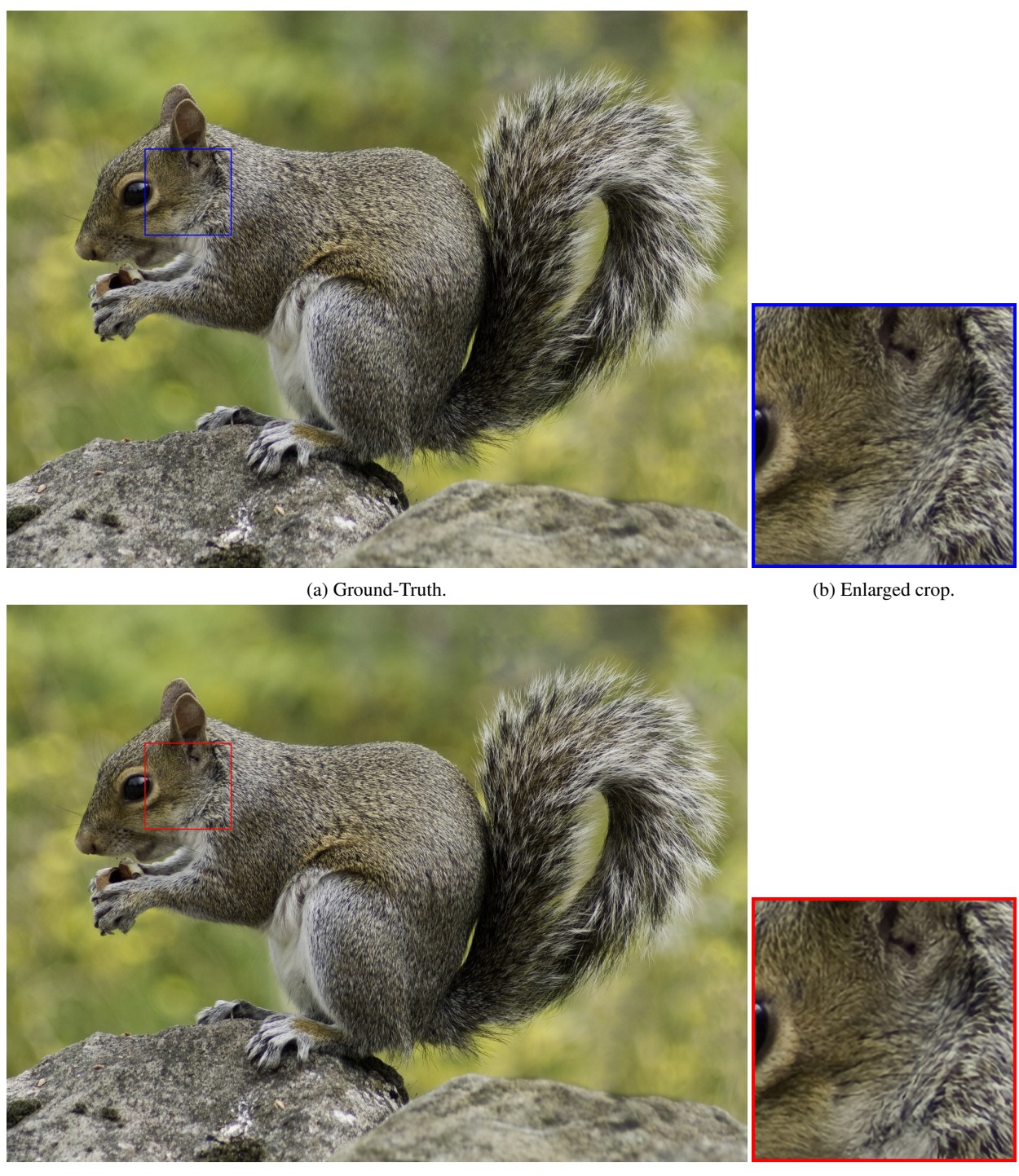

(a) Ground-Truth.

(b) Enlarged crop.

(c) Enlarged crop.

(d) Enlarged crop.

Figure 6: Ground-truth high resolution image (top) and the output of our Boolean super-resolution methodology (bottom). Image "0810" from the validation set of DIV2k, with PSNR: 34.90 dB

### B.7.2. TRAINING SETUP

The model was trained on the Cityscapes dataset for $400$ epochs with a batch size of $8$. The AdamW optimizer (Loshchilov & Hutter, 2017) with an initial learning rate of $5 \times 10^{-4}$ and the Boolean logic optimizer with a learning rate of $12$ were

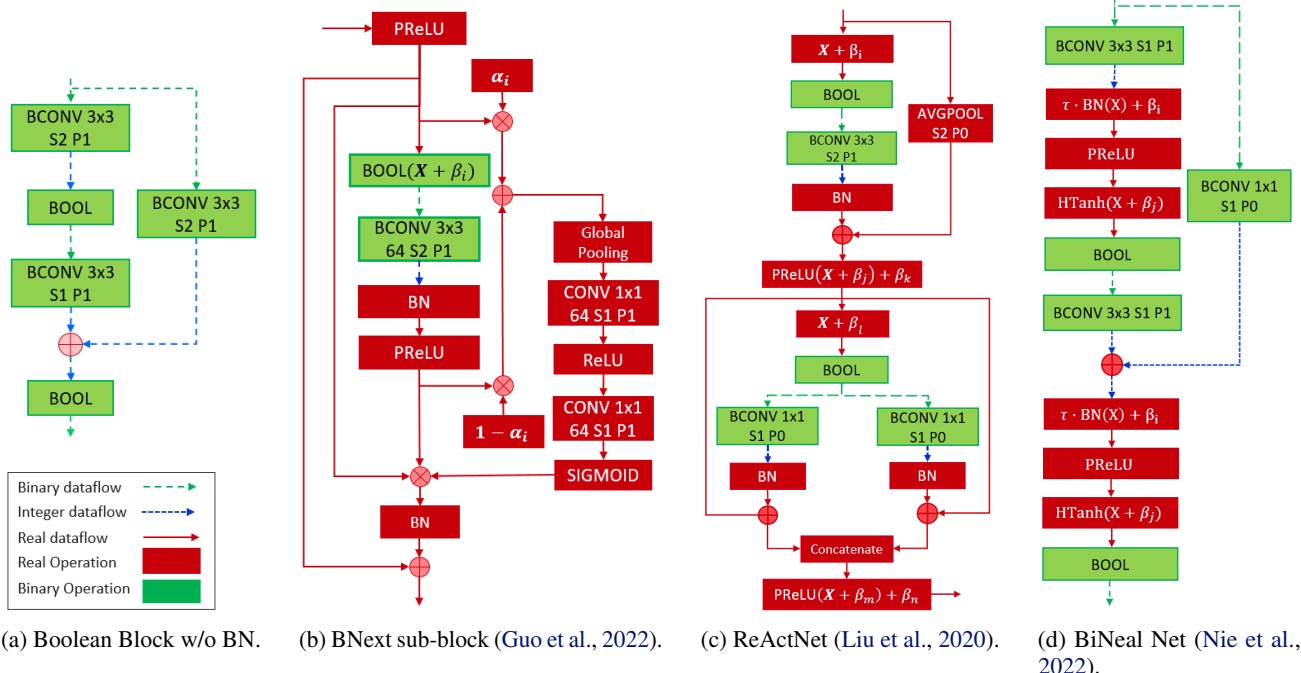

(a) Boolean Block w/o BN.  (b) BNext sub-block (Guo et al., 2022).  (c) ReActNet (Liu et al., 2020).  (d) BiNeal Net (Nie et al., 2022).

Figure 7: Comparative graph of popular BNN techniques and our Boolean module. Notice how multiple full-precision operations like BN, PReLU, or Squeeze-and-Excitation are overly used on each BNN block.

used respectively for real and Boolean parameters. At the early training stage, parameters could easily be flipped due to the large backward signal; thus, to better benefit from the ImageNet-pretrained backbone, we reduce the learning rate for Boolean parameters in the backbone to 6. We employed the polynomial learning rate policy with $p = 0.9$ for all parameters. The cross-entropy loss was used for optimization. We did not employ auxiliary loss or knowledge distillation as these training techniques require additional computational cost, which is not in line with our efficient on-device training objective.

### B.7.3. DATA SAMPLING AND AUGMENTATION

We aim to reproduce closely full-precision model performance in the semantic segmentation task with Boolean architecture and Boolean logic training. Due to the nature of the Boolean network, the common regularization method, e.g., weight decay, is not applicable. Moreover, with more trainable parameters, the Boolean network can suffer from over-fitting. In particular, as shown in Table 13, the imbalanced dataset for semantic segmentation aggravates the situation. There is a significant performance gap for several classes which has low occurrence rate, including *rider (9.5%), motor (11.2%), bus (9.5%), truck (6.9%), train (17.0%)*. We argue that the performance gap is due to the similarity between classes and the dataset's low occurrence rate, which is confirmed as shown in Figure 10.

Data augmentation and sampling are thus critical for Boolean model training. Regarding data augmentation, we employed multi-scale scaling with a random scaling factor ranging from 0.5 to 2. We adopted a random horizontal flip with probability $p = 0.5$ and color jittering. In addition, we used rare class sampling (RCS) (Hoyer et al., 2022) to avoid the model over-fitting to frequent classes. For class $c$, the occurrence frequency in image $f_c$ is given by:

$$f_c = \frac{\sum_{i=1}^{N} \mathbf{1}(c \in y_i)}{N}, \tag{9}$$

where $N$ is the number of samples and $y_i$ is the set of classes existing in sample $i$. The sampling probability of class $c$ is thus defined as:

$$p_c = \frac{\exp\left(\frac{1-f_c}{T}\right)}{\sum_{c'=1}^{K} \exp\left(\frac{1-f_{c'}}{T}\right)}, \tag{10}$$

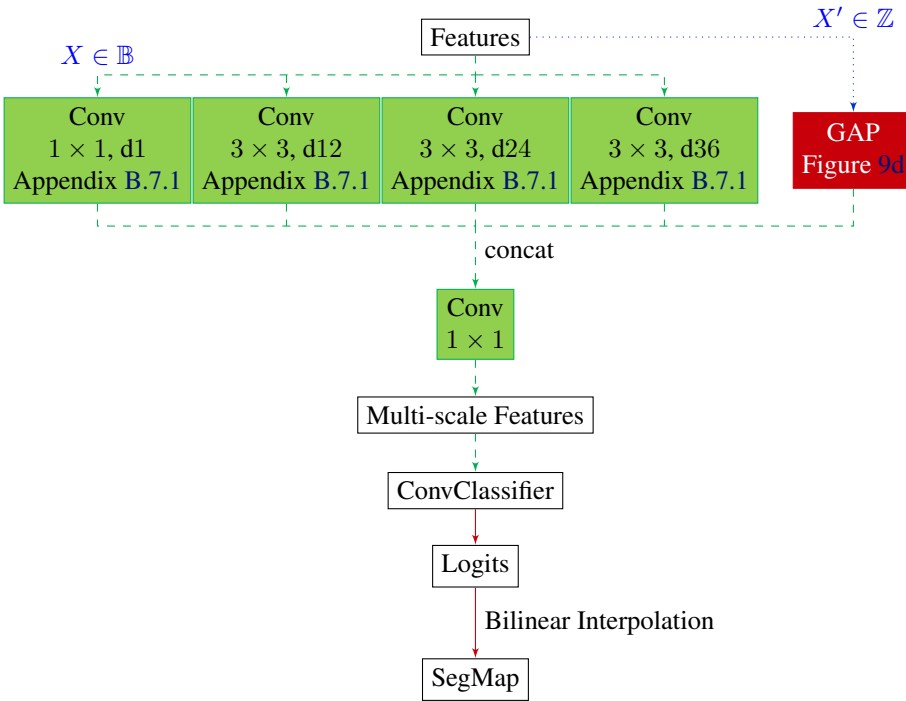

Figure 8: Boolean segmentation architecture.

|  | Image ratio (%) | Δ mIoU (%) |
|---|---|---|
| Road† | 98.62 | 0.0 |
| Sideway† | 94.49 | 0.7 |
| Building† | 98.62 | 0.6 |
| Wall | 32.61 | 7.4 |
| Fence | 43.56 | 3.8 |
| Pole | 99.13 | 1.8 |
| Light | 55.73 | 6.5 |
| Sign† | 94.39 | 2.8 |
| Vegetation† | 97.18 | 0.1 |
| Terrain† | 55.60 | 0.8 |
| Sky† | 90.29 | −0.2 |
| Person | 78.76 | 1.5 |
| Rider* | 34.39 | 7.4 |
| Car† | 95.19 | 0.3 |
| Truck* | 12.07 | 6.9 |
| Bus* | 9.21 | 12.8 |
| Train* | 4.77 | 17.0 |
| Motor* | 17.24 | 15.3 |
| bike | 55.33 | 2.0 |

Table 13: Class per image and performance gap occurrence rates in Cityscapes training set with naive Bool ASPP design. Class with low performance gap† and class with high performance gap*.

where $K$ is the number of classes, and $T$ is a hyper-parameter for sampling rate balancing. In particular, for the Cityscapes dataset, we selected $T = 0.5$.

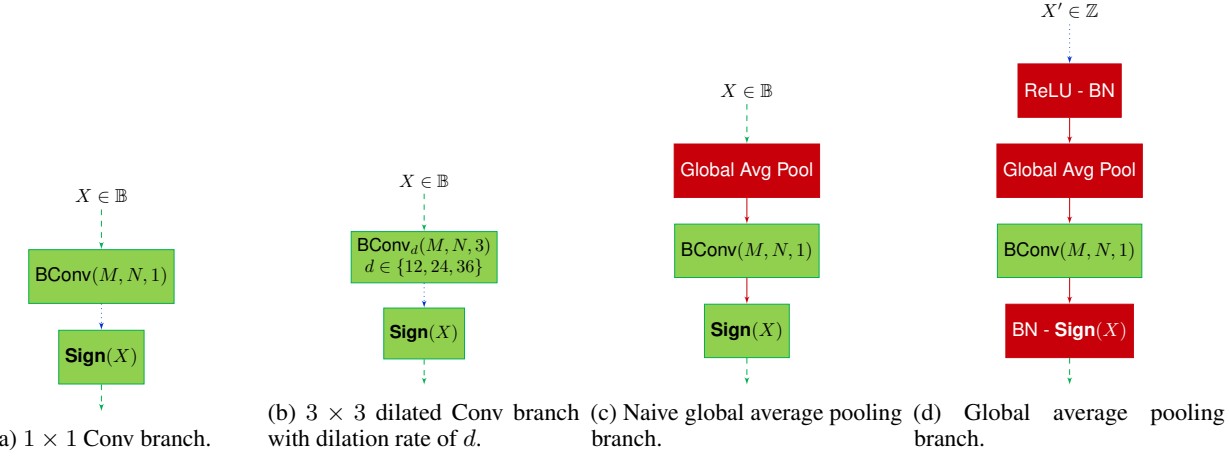

Figure 9: Boolean Atrous Spatial Pyramid Pooling (BoolASPP) architecture.

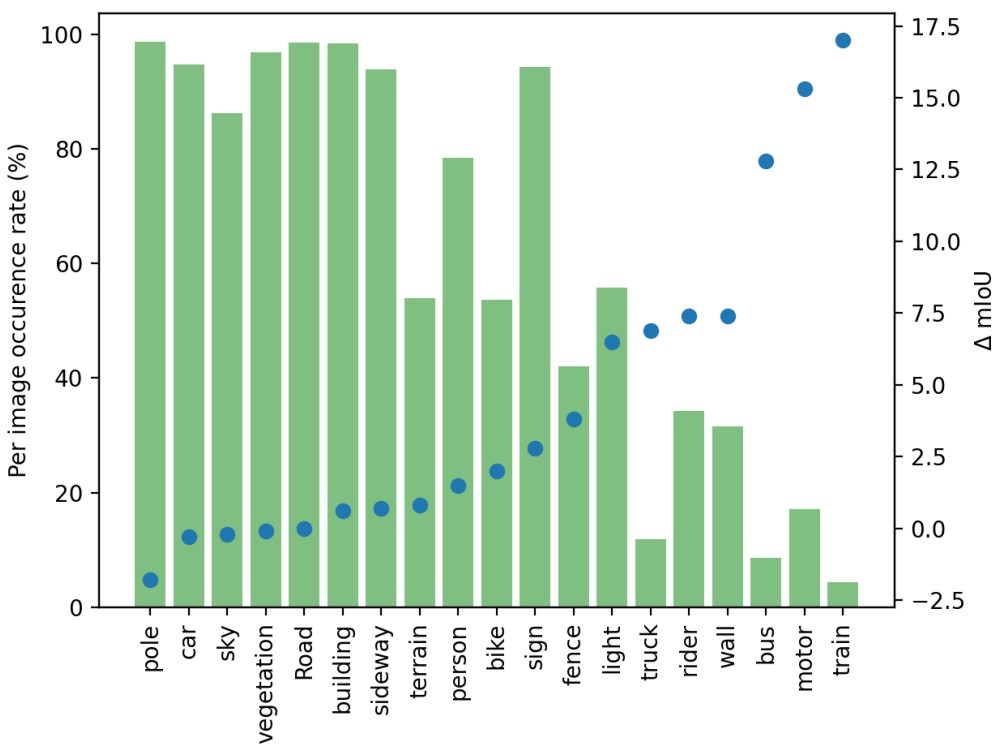

Figure 10: Class per image occurrence ratio and performance gap with naive Bool ASPP design.

### B.7.4. QUALITATIVE ANALYSIS ON CITYSCAPES VALIDATION SET

The qualitative results of our Boolean network and the full-precision based are demonstrated in Figure 11. Despite the loss of model capacity, the proposed Boolean network trained with Boolean logic optimizer has comparable performance with large objects in the frequent classes, even in the complicated scene.

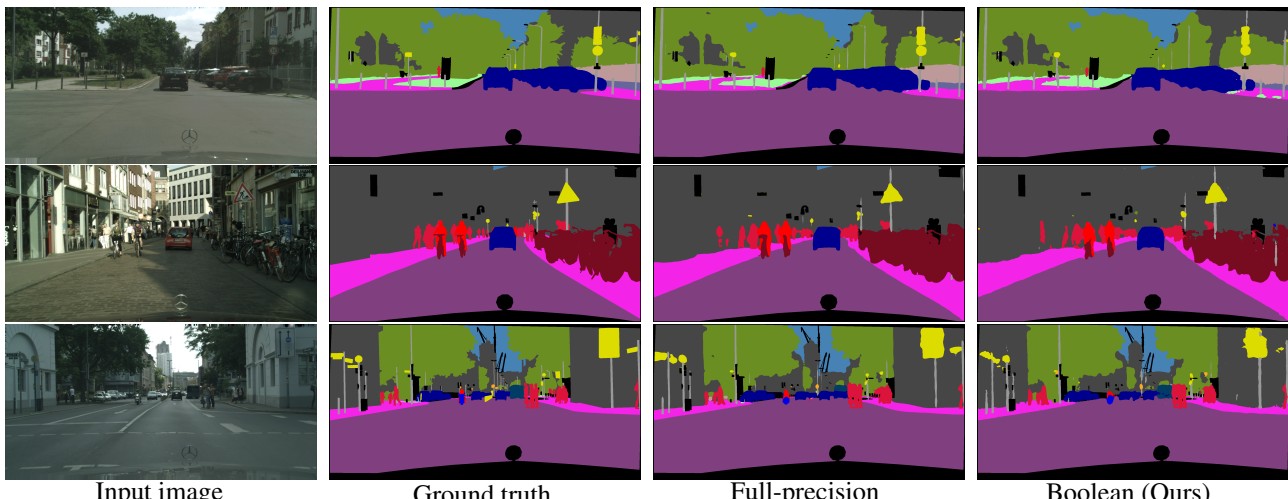

| Input image | Ground truth | Full-precision | Boolean (Ours) |

Figure 11: Visual comparison of Boolean model on Cityscapes validation set.

| Seg. head | Model | mIoU (%) | Δ |
|---|---|---|---|
| FCN-32s (Long et al., 2015) | FP baseline | 64.9 | - |
| | Group-Net (Zhuang et al., 2019) | 60.5 | 4.4 |
| | Ours | 60.1 | 4.8 |
| DeepLabv3 (Chen et al., 2017) | FP baseline | 72.1 | - |
| | Ours | **67.3** | 4.8 |

Table 14: Performance on Pascal VOC 2012 val set.

| Methods | road | sideway | building | wall | fence | pole | light | sign | vegetation | terrain | sky | person | rider | car | truck | bus | train | motor | bike | mIoU |
|---|---|---|---|---|---|---|---|---|---|---|---|---|---|---|---|---|---|---|---|---|
| FP baseline | 97.3 | 79.8 | 90.1 | 48.5 | 55.0 | 49.4 | 59.2 | 69. | 90.0 | 57.5 | 92.4 | 74.3 | 54.6 | 91.2 | 61.4 | 78.3 | 66.6 | 58.0 | 70.8 | 70.7 |
| Naive Bool ASPP | 97.3 | 79.1 | 89.5 | 41.1 | 51.2 | 51.2 | 52.7 | 66.2 | 90.1 | 56.7 | 92.6 | 72.8 | 47.2 | 91.5 | 54.5 | 65.5 | 49.6 | 42.7 | 68.8 | 66.3 |
| Δ | **0.0** | 0.7 | 0.6 | 7.4 | 3.8 | -1.8 | 6.5 | 2.8 | -0.1 | 0.8 | -0.2 | **1.5** | **7.4** | -0.3 | 6.9 | 12.8 | **17.0** | 15.3 | 2.0 | 4.4 |
| Ours | 97.1 | 78. | 89.8 | 46.2 | 51.3 | 52.7 | 53.3 | 66.5 | 90.2 | 58. | 92.7 | 72.6 | 45.1 | 91.9 | 61.1 | 68.8 | 48.7 | 46.8 | 69.1 | 67.4 |
| Δ | 0.2 | 1.8 | **0.3** | 2.3 | 3.7 | -3.3 | 5.9 | 2.5 | -0.2 | -0.5 | -0.3 | 1.7 | 9.5 | **-0.7** | 0.3 | 9.5 | 17.9 | **11.2** | 1.7 | 3.3 |

Table 15: Class-wise IoU performance on Cityscapes validation set.

### B.7.5. MORE EXPERIMENTS ON SEMANTIC SEGMENTATION

We evaluated the effectiveness of BoolASPP by investigating the per-class performance gap to the full-precision model. As demonstrated in Table 15, a significant gap exists between the Boolean architecture with naive Boolean ASPP design; i.e., using Boolean activations for ASPP module as illustrated in Figure 9c. However, the gap could be reduced by using BoolASPP and RCS. In particular, the BoolASPP improves the IoU of *truck* from 54.5% to 64.1% and *bus* from 65.5% to 68.8%, *bike* from 68.8% to 69.1% and *motor* from 42.8% to 46.8%. This indicates that combining proposed BoolASPP and RCS improves the model performance on low occurrence classes as well as similar classes with which are easy to be confused.

### B.7.6. VALIDATION ON PASCAL VOC 2012 DATASET

We also evaluated our Boolean model on the 21-class Pascal VOC 2012 dataset with augmented additional annotated data containing 10, 582, 1, 449, and 1, 456 images in training, validation, and test set, respectively. The same setting is used as in

the experiments on the Cityscapes dataset, except the model was trained for $60$ epochs.

As shown in Table 14, our model with fully Boolean logic training paradigm, i.e., without any additional intermediate latent weight, achieved comparable performance as the state-of-the-art latent-weight-based method. Our Boolean model improved performance by incorporating multi-resolution feature extraction modules to $67.3\%$ mIoU.

