# OpenReview forum: "Boolean Logic for Low-Energy Deep Learning"
_ICML.cc/2024/Workshop/WANT — WANT@ICML 2024 Poster_

### Official Review · Reviewer_821g · 2024-06-11
**Boolean Logic for Low-Energy Deep Learning**

**Confidence:** 3

**Summary:**

This paper presents an approach to evaluate the energy in both training and inference for neural architectures using Boolean logic backpropagation. The use of Boolean logic was a central idea in studying the model's efficiency. It's interesting to see its benefits for memory energy. A few suggestions were given to improve the paper.

**Strengths:**

The state of the art is well-written and the experiments for fine-tuning and super-resolution are encouraging and well-explained.

**Weaknesses:**

- The paper doesn't explicitly explain the computation of energy cost for data transfer. Particularly, equation 4 (line 265) seems intuitive but not proved rigorously, also, I could not relate that to the cited paper. Maybe you can explain the intuition behind the energy cost of moving data to DRAM. I have the same remark for equation 5.
- ADD-INT in section 3.2.1 (line 176) was never explained before.
- References do not appear in some sections (lines 252, 739, 805).
- your model doesn't show a significant gain in energy compared to baselines for large models for ImageNet classification.

**Limitations:**

The paper states that Boolean networks replace the complex calculations of gradients during the Backpropagation. However, replacing the classic approach requires more details about differential calculus, namely for binary models. This was used in equations 6 and 7, without explaining assumptions for backpropagation. The authors wrote a short sentence in Appendix A.2, but it doesn't explain gradient computation in the existence of residual boolean blocks.

**Suggestions:**

- I'd suggest that this subsection 3.2 requires more effort to state the problem instead of using references, the related work for energy estimation can go to related work.
- The number of times filters and ofmaps are reused or the frequency of access to each memory level may also be a key factor in designing a new adaptive approach.

---

### Official Review · Reviewer_91nC · 2024-06-11
**Promising Boolean neuron design and Boolean logic backpropagation to replace traditional gradient descent**

**Confidence:** 4

**Summary:**

The paper "Boolean Logic for Low-Energy Deep Learning" introduces an innovative approach by utilizing Boolean neuron design and Boolean logic backpropagation, replacing traditional gradient descent and real arithmetic. This method significantly reduces energy consumption, making it ideal for edge devices with limited computational resources. Comprehensive evaluation, including detailed energy estimations for both training and inference phases, reinforces the credibility of the results. The method is validated across various tasks such as image classification, fine-tuning, and image super-resolution, showcasing its versatility. Extensive supplementary materials in the appendix provide further insights into the work. However, energy savings are partially hardware-dependent, and the paper would benefit from a more thorough discussion of limitations and potential challenges in different scenarios.

**Strengths:**

The introduction of Boolean neuron design and Boolean logic backpropagation presents a novel and innovative approach, challenging the traditional reliance on gradient descent and real arithmetic.

The paper delivers a detailed and comprehensive evaluation of the proposed method, including an energy estimation for both the training and inference phases. This in-depth analysis reinforces the credibility of the results.

The method has been validated across a variety of tasks, such as image classification, fine-tuning, and image super-resolution, showcasing its adaptability and resilience.

For practical applications, especially in fine-tuning large models on edge devices with constrained computational resources, the method demonstrates promising potential, underscoring its real-world relevance.

The additional materials provided in the appendix are extensive and offer further insight into the work presented.

**Weaknesses:**

Energy savings are partially reliant on particular hardware architectures, like the Ascend chip architecture utilized in the experiments. Performance and energy efficiency may differ across various hardware platforms.

The paper would be enhanced by a more comprehensive examination of the approach's limitations and potential challenges, including situations where it may not perform as well.

---

### Official Review · Reviewer_BhcX · 2024-06-13
**Interesting fully binary network for training and inference with good performance and low energy consumption**

**Confidence:** 4

**Summary:**

The paper proposes a study of binary neural networks using Boolean logic training together with optional knowledge distillation. They show that on some tasks they can achieve results close to a full-precision network with vastly reduced energy consumption. These tasks include super-resolution and segmentation, unlike previous results on BN that focus on classification.

**Strengths:**

The paper is clear and well written. It is also well on topic for this workshop. The method of Boolean logic back propagation is very interesting and novel to this reviewer, although published earlier in Nguyen 2023.


The results are very good and encouraging.

**Weaknesses:**

It is not clear why authors focus on small networks.

**Limitations:**

What is the time efficiency of boolean logic back propagation?

---

### Meta-Review · Area_Chair_aBMd · 2024-06-17

**Recommendation:** Accept (Poster)
**Confidence:** 5

**Metareview:**

The paper introduces a new approach for training neural networks efficiently using boolean logic, which is hardware friendly and energy efficient. Overall, it's an interesting read and a promising direction for energy savings. All 3 reviews received are generally positive. The authors should try to address the comments received  by the camera ready.

---

### Decision · Program_Chairs · 2024-06-17

**Decision:**

Accept (Poster)

**Comment:**

We thank the authors for their time and contribution to WANT and we are pleased to share that after the reviewing process the paper has been accepted. Congratulations! We encourage the authors to consider reviewers' feedback for the improvement of the camera-ready version. We hope to see you in person at the workshop and brainstorm on efficient training research together!